# Impact of *Bradyrhizobium elkanii* and *Azospirillum brasilense* Co-Inoculation on Nitrogen Metabolism, Nutrient Uptake, and Soil Fertility Indicators in *Phaseolus lunatus* Genotypes

**DOI:** 10.3390/plants15010135

**Published:** 2026-01-02

**Authors:** Gislayne Kayne Gomes da Cruz, José Aliff da Silva de Souza, José Félix de Brito Neto, Cristiano dos Santos Sousa, Samara Lima Brito, Maria Geovana Martins Souza, Evandro Franklin de Mesquita, Rodrigo Santana Macedo, Raíres Liliane de Oliveira Cruz, Vicente Victor Lima de Andrade, Walter Esfrain Pereira, Rennan Fernandes Pereira

**Affiliations:** 1Postgraduate Program in Agricultural Sciences, Paraíba State University, Campina Grande 58429-500, PB, Brazil; gislayne.kayne.gomes.cruz@aluno.uepb.edu.br (G.K.G.d.C.); felix.brito@servidor.uepb.edu.br (J.F.d.B.N.); cs.solos2023@gmail.com (C.d.S.S.); samaralimab@gmail.com (S.L.B.); evandrofranklin@servidor.uepb.edu.br (E.F.d.M.); 2Department of Agroecology, Federal Institute of Paraíba, Picuí 58187-000, PB, Brazil; aliff.rozeno@academico.ifpb.edu.br; 3Department of Agricultural and Environmental Sciences, Paraíba State University, Lagoa Seca 58117-000, PB, Brazil; souza.geovana@aluno.uepb.edu.br (M.G.M.S.); vicente.aluno@gmail.com (V.V.L.d.A.); 4Academic Unit of Agricultural Sciences, Federal University of Campina Grande, Pombal 58840-000, PB, Brazil; rodrigo.santana@professor.ufcg.edu.br; 5Postgraduate Program in Agricultural Engineering, Federal University of Campina Grande, Campina Grande 58429-900, PB, Brazil; rairescrux@gmail.com; 6Postgraduate Program in Agronomy, Federal University of Paraíba, Areia 58397-000, PB, Brazil; walterufpb@yahoo.com.br

**Keywords:** biological nitrogen fixation, lima bean, plant growth-promoting rhizobacteria, root nodulation, pulses

## Abstract

Lima bean (*Phaseolus lunatus* L.), an important legume in semiarid environments, often exhibits low yield, requiring strategies to enhance symbiotic nitrogen fixation and nutrient-use efficiency. This study evaluated the effects of single and combined inoculation with *Bradyrhizobium elkanii* (strain BR 2003) and *Azospirillum brasilense* (strain Ab-V5) on nitrogen metabolism, nutrient uptake, plant growth, and residual soil fertility in *P. lunatus*. Four varieties were subjected to four treatments: control (nitrogen fertilization), single inoculation with *B. elkanii* or *A. brasilense*, and co-inoculation. All inoculation strategies significantly increased root nodulation, nitrogen assimilation, and the accumulation of key macronutrients. Root nodulation increased from 1 to 12 nodules per plant in the control treatments to up to 277 nodules per plant under inoculation, while shoot nitrogen content increased by up to 91% in ‘Raio de Sol’ and 87% in ‘Cearense’. Increases in P and K were also observed, including a 48% increase in shoot P in ‘Manteiga’ and up to a 100% increase in shoot K in ‘Raio de Sol’, whereas root K increased by up to 90% under co-inoculation. The ‘Raio de Sol’ and ‘Manteiga’ varieties exhibited the most pronounced increases in growth and biomass. Additionally, inoculation improved post-cultivation soil indicators, including pH and available P and K in specific genotype-microbe combinations, and reduced electrical conductivity. These results demonstrate the strong contribution of microbial inoculation to nitrogen assimilation and nutrient acquisition, supporting its use as a promising alternative to conventional nitrogen fertilization in lima bean cultivation.

## 1. Introduction

Legumes are key components of the global food system due to their high protein content, essential amino acids, fiber, and micronutrients, which make them vital for human nutrition and food security, especially in developing countries [1,2]. According to the Food and Agriculture Organization of the United Nations (FAO), global consumption of dried legumes (pulses) reached 101 million tons in 2024, with production mainly concentrated in India, Canada, China, and the European Union [3], reinforcing their global economic relevance. FAO per capita consumption data indicate that Africa, Latin America and the Caribbean are the regions with the highest consumption of pulses, reflecting their strong cultural, dietary, and socioeconomic importance [3].

Beyond their nutritional value, legumes also play a central role in agricultural systems due to their ability to establish symbiotic associations with nitrogen-fixing bacteria, reducing dependence on mineral fertilizers and contributing to soil fertility [4,5]. However, the efficiency of biological nitrogen fixation (BNF) varies considerably among species, genotypes, and edaphoclimatic conditions, and remains insufficiently explored with respect to strategies aimed at enhancing BNF through interactions with plant growth-promoting rhizobacteria [6,7].

Lima bean (*Phaseolus lunatus* L.), a member of the Fabaceae family, stands out among grain legumes due to its high nutritional value and wide adaptability to tropical environments [8]. Although it belongs to the same genus as *Phaseolus vulgaris*, lima beans exhibit distinct genetic and physiological traits, including greater germplasm variability and differences in root architecture and flavonoid-mediated signaling, which may influence nodulation capacity and BNF efficiency [9,10].

Recent studies indicate that *P. lunatus* displays high variability in symbiotic performance, often showing reduced nodulation stability compared with crops such as soybean and common bean [11,12]. Nevertheless, its seeds contain substantial amounts of protein, complex carbohydrates, essential amino acids, and key micronutrients, reinforcing its relevance as a food resource in diverse regions [13]. This nutritional profile is particularly important in semi-arid areas, where the crop contributes significantly to food security and local income generation.

Despite this potential, yield of lima bean in Brazil remains low, with an average of 329 kg ha^−1^ in 2024 [14]. This limitation is often associated with low nodulation efficiency, reduced lateral root density compared with common bean, and low availability of nitrogen and organic matter in soils, as well as sensitivity to low fertility, soil acidity, and water stress conditions [15,16,17]. Although *P. lunatus* can establish symbiotic relationships with diazotrophic bacteria for BNF, this potential is often underexplored due to the absence of specific inoculants and variability in nodular response between varieties [18,19,20].

Varieties such as ‘Raio de Sol’, ‘Manteiga’, ‘Cearense’, and ‘Orelha-de-Vó’ represent a wide diversity of crops originating from different micro-regions, characterized by distinct morphological traits and adaptation patterns, thereby constituting an appropriate model for investigating genotype-dependent microbial responses [9]. The literature indicates that lima bean production in Brazil is mainly concentrated in the states of Pernambuco, Paraíba, Ceará, Rio Grande do Norte, Alagoas, Bahia, and Piauí, where these seeds are maintained as creole varieties locally selected and conserved by smallholder farmers [14].

The ‘Manteiga’ and ‘Raio de Sol’ varieties are predominantly cultivated in the states of Pernambuco and Paraíba, whereas the ‘Cearense’ variety is mainly found in the Cariri and Sertão regions of Ceará, reflecting its adaptation to semi-arid climatic conditions [21]. In contrast, the ‘Orelha-de-Vó’ variety predominates in rural communities of Pernambuco, Paraíba, and Rio Grande do Norte, where it is associated with the conservation of traditional seeds and the adoption of agroecological practices by local farming communities [22]. This marked regional distribution highlights the high genetic diversity and strong cultural significance of lima bean cultivation in Northeastern Brazil, reinforcing its value as a biological model for studies addressing genotype-dependent microbial interactions and adaptive responses to contrasting environmental conditions.

*P. lunatus* exhibits considerable intraspecific variability in nodule formation and functionality, influenced by genetic factors that regulate root system development, signaling pathways, and compatibility with symbiotic bacteria [23]. Genotypic differences are evident in nodulation responses, with ‘Manteiga’ and ‘Raio de Sol’ generally showing higher nodulation and BNF stability under medium-fertility soils, whereas ‘Cearense’ and ‘Orelha-de-Vó’, commonly cultivated in semi-arid environments, tend to exhibit delayed nodulation or reduced nodule density, likely due to physiological traits associated with abiotic stress tolerance [9,21]. Such genotype-dependent variability highlights the need for management strategies capable of enhancing symbiotic efficiency across contrasting genetic backgrounds and environmental conditions.

In this context, co-inoculation with plant growth-promoting rhizobacteria (PGPR) has emerged as an effective approach to improve plant nodulation, nitrogen metabolism, nutrient uptake, and growth, while reducing reliance on chemical fertilizers and improving soil quality [24,25,26,27]. The efficiency of PGPR is strongly influenced by the compatibility between plant genotype and microorganism, as well as environmental conditions, leading to variable responses between varieties and cultivation sites [24,25].

Among the most studied PGPRs, *Bradyrhizobium elkanii* is widely recognized for its role in symbiotic nitrogen fixation, while *Azospirillum brasilense* promotes root growth and nutrient uptake through the synthesis of phytohormones such as indole-3-acetic acid (IAA) and abscisic acid (ABA), as well as phosphate solubilization and siderophore production [28,29,30,31,32,33]. Studies in soybean, common bean, and cowpea have shown that co-inoculation with *Rhizobium* and *Azospirillum* species can produce synergistic effects, increasing nodulation, biomass accumulation, and tolerance to abiotic stresses [32,33,34,35,36]. These responses are commonly attributed to increased availability of assimilable nitrogen, greater root surface area, and modulation of biochemical processes related to nutrient assimilation and energy metabolism [37,38,39]. Similar synergistic mechanisms have also been reported in cowpea, where co-inoculation resulted in increased nitrogen accumulation, vegetative growth, and productivity without compromising symbiotic efficiency [40].

Despite this evidence in other legumes, the potential benefits of co-inoculating PGPR in *P. lunatus* remain poorly explored. Given the physiological and symbiotic particularities of this species, it is essential to understand how different inoculation strategies (single or combined) influence nodulation, nitrogen metabolism, nutrient uptake, and soil quality parameters. An integrated evaluation of these factors enables the investigation of plant–soil feedback mechanisms and the potential of microbial practices to enhance physiological performance in the medium and long term, as well as to verify whether the synergistic effects observed in other legumes are consistent in lima bean.

From this perspective, investigating co-inoculation strategies in lima bean represents a relevant and innovative research gap, as most existing studies focus on common bean and cowpea. Therefore, this study aimed to evaluate the effects of simple and combined inoculation with *B. elkanii* and *A. brasilense* strains on root nodulation, nitrogen metabolism, macronutrient uptake and morphological characteristics in different varieties of lima bean (*P. lunatus*), as well as on soil fertility indicators after cultivation. We hypothesized that (i) inoculation strategies with *B. elkanii* and *A. brasilense* increase the growth and biomass of *P. lunatus* (nitrogen uptake); and (ii) microbial inoculation increases the soil chemical quality.

## 2. Results

### 2.1. Root Nodulation

Factorial ANOVA revealed significant effects of both variety and inoculation treatment on nodule number and nodule dry mass. All inoculation treatments significantly increased both parameters compared with the non-inoculated control (Figure 1). In the control, the mean numbers of nodules per plant were 12, 2, 1, and 1 for ‘Raio de Sol’, ‘Manteiga’, ‘Orelha-de-vó’, and ‘Cearense’, respectively (Figure 1a). Inoculation with *B. elkanii*, *A. brasilense*, or their combination resulted in statistically significant increases in nodule biomass and nodule number across all varieties compared with the non-inoculated treatment. The varieties ‘Raio de Sol’, ‘Manteiga’, and ‘Orelha-de-vó’ produced averages of 277, 154, and 195 nodules, respectively, with no significant differences among inoculation treatments. In ‘Cearense’, the mean number of nodules under all inoculated conditions was 147. Nodule dry mass followed the same trend, showing pronounced increases for all inoculated plants (Figure 1b).

Representative root samples of the ‘Raio de Sol’ variety are shown in Figure 2. Roots from the non-inoculated control (Figure 2a) and those inoculated with *B. elkanii* (Figure 2b) illustrate the difference in nodulation intensity, confirming the quantitative results of Figure 1a. Inoculated plants developed a markedly higher number of nodules, which were distributed along both primary and lateral roots, predominantly spherical, smooth-surfaced, and light brown externally. When sectioned, nodules exhibited a pink interior, indicative of active nitrogen fixation.

### 2.2. Nitrogen Accumulation in Different Plant Organs

Factorial ANOVA showed that the interaction between variety and inoculation treatment significantly affected nitrogen contents in shoots, roots, and nodules, as well as NH_4_^+^ and NO_3_^−^ concentrations in shoots. In the ‘Raio de Sol’ variety, plants inoculated with *B. elkanii* or *A. brasilense* exhibited shoot nitrogen contents 84% and 91% higher than the control, respectively (Figure 3a). Similarly, in ‘Cearense’, the same treatments increased shoot nitrogen contents by 79% and 87%. Co-inoculation significantly increased shoot nitrogen only in the ‘Orelha-de-vó’ variety, by 77% relative to the control, while none of the inoculation treatments affected shoot nitrogen in ‘Manteiga’.

In roots, nitrogen contents in ‘Raio de Sol’ and ‘Orelha-de-vó’ did not differ significantly among treatments (Figure 3b). In ‘Cearense’, *B. elkanii* inoculation increased root nitrogen content by 78% compared with the control, whereas other treatments showed no differences. In contrast, in ‘Manteiga’, root nitrogen contents under *A. brasilense* inoculation alone and co-inoculation were 32% and 29% lower, respectively, than the control.

Nodule nitrogen contents were higher in all inoculated plants compared with the control (Figure 3c). In ‘Raio de Sol’, the control treatment showed 0.7 g kg^−1^, while in ‘Manteiga’, ‘Orelha-de-vó’, and ‘Cearense’ controls, no nodule nitrogen was detected. In ‘Raio de Sol’, all inoculated treatments increased nodule nitrogen by an average of 221%, reaching 2.25 g N kg^−1^ dry mass. Across the inoculated treatments of ‘Raio de Sol’, ‘Orelha-de-vó’, and ‘Cearense’, nitrogen contents ranged from 1.07 to 2.56 g N kg^−1^, with no significant differences among bacterial treatments.

Regarding inorganic nitrogen forms, only the ‘Manteiga’ variety showed no significant differences in shoot NH_4_^+^ (Figure 4a) and NO_3_^−^ (Figure 4b) contents among treatments. In the other varieties, inoculation enhanced both variables, although the magnitude varied by genotype. In ‘Raio de Sol’, *A. brasilense* and *B. elkanii* individually promoted the highest NH_4_^+^ and NO_3_^−^ levels, increasing them by 92% and 84% relative to the control. In ‘Orelha-de-vó’, co-inoculation produced the highest ammonium and nitrate concentrations, both increasing by 77%. In ‘Cearense’, all inoculated treatments enhanced NH_4_^+^ and NO_3_^−^ contents by an average of 81% over the control.

### 2.3. Nutrient Contents in Different Plant Organs

Macronutrient accumulation in lima bean plants was significantly influenced by inoculation treatments and varied among cultivars. Factorial ANOVA revealed significant effects of treatment on phosphorus, potassium, calcium, and magnesium concentrations in shoots, roots, and nodules.

Shoot phosphorus (Figure 5a) responses varied among cultivars. In ‘Raio de Sol’, no significant differences were detected between inoculated and control plants. In ‘Manteiga’, inoculation with *B. elkanii* increased shoot P by 48% relative to the control. All inoculation treatments enhanced shoot P in ‘Orelha-de-vó’ by an average of 61%, with no significant differences among treatments. In ‘Cearense’, only co-inoculation significantly increased shoot P, by 26% compared with the control. Root P (Figure 5b) was significantly higher following *B. elkanii* inoculation in ‘Raio de Sol’ and ‘Orelha-de-vó’, with increases of 58% and 54%, respectively. In ‘Orelha-de-vó’, *A. brasilense* also increased root P by 46% compared with the control.

In ‘Manteiga’, ‘Orelha-de-vó’, and ‘Cearense’, the extremely low nodulation observed in non-inoculated controls (Figure 1a) resulted in negligible P (Figure 5c), K (Figure 6c), Ca (Figure 7c), and Mg (Figure 8c) contents in nodules. In contrast, all inoculated plants exhibited marked increases in nodule macronutrient contents, with ‘Raio de Sol’ showing the highest overall values. In this variety, which also presented the greatest number of nodules among control plants, nodule P content increased from 0.56 mg kg^−1^ in the control to 2.28 mg P kg^−1^ dry mass following *B. elkanii* inoculation, a 307% increase.

Shoot potassium (Figure 6a) generally increased under inoculation. In ‘Raio de Sol’ and ‘Manteiga’, inoculation with *B. elkanii*, *A. brasilense*, or their combination produced similar shoot K concentrations, averaging 100% and 48% higher than the control, respectively. In ‘Orelha-de-vó’ and ‘Cearense’, *A. brasilense* alone did not differ from the control, whereas *B. elkanii* and co-inoculation increased shoot K by 38% and 31%, respectively. Root potassium (Figure 6b) also rose significantly in co-inoculated plants, with increases of 90% in ‘Raio de Sol’ and 70% in ‘Manteiga’. In ‘Orelha-de-vó’, *A. brasilense* increased root K by 112%, while in ‘Cearense’, all inoculated treatments showed similar root K levels, averaging 45% higher than the control.

Shoot calcium (Figure 7a) exhibited cultivar-specific patterns. No significant differences were observed in ‘Raio de Sol’ or ‘Manteiga’. In ‘Orelha-de-vó’, *A. brasilense* increased shoot Ca by 50%. In ‘Cearense’, shoot Ca contents were similar among control, *A. brasilense*, and co-inoculated plants, whereas *B. elkanii* reduced shoot Ca by 32% compared with the control. Root Ca (Figure 7b) largely mirrored these trends. Treatments did not differ in ‘Raio de Sol’ or ‘Manteiga’. In ‘Orelha-de-vó’, all inoculations produced similar root Ca, increasing it by approximately 660% compared with the control. In ‘Cearense’, root Ca was highest in the control and statistically similar to *B. elkanii*; *A. brasilense* and co-inoculation produced lower, though still comparable, levels to the higher values observed in other cultivars.

Magnesium accumulation showed clear organ- and cultivar-specific trends. Foliar Mg (Figure 8a) was not enhanced by inoculation in any cultivar and was 37% lower than the control in ‘Manteiga’. Root Mg (Figure 8b) remained unaffected in ‘Raio de Sol’, ‘Manteiga’, and ‘Cearense’, but in ‘Orelha-de-vó’, *A. brasilense* increased root Mg by 48%.

### 2.4. Plant Growth and Biomass Accumulation

Growth and biomass traits were significantly influenced by the interaction between lima bean varieties and inoculation treatments, as revealed by factorial ANOVA. In ‘Raio de Sol’, inoculation with *B. elkanii* increased plant height by 70% compared with the control (Figure 9a). The ‘Manteiga’ variety exhibited the most pronounced response, with *A. brasilense*, *B. elkanii*, and co-inoculation increasing plant height by 707%, 453%, and 440%, respectively. Conversely, ‘Orelha-de-vó’ displayed an opposite trend, with the control producing the tallest plants, although co-inoculated plants did not differ significantly from the control. In ‘Cearense’, plant height was similar among the control, *A. brasilense*, and co-inoculated treatments, while *B. elkanii* inoculation alone resulted in the lowest values.

Dry matter accumulation across organs followed the same general trend as plant height. In both ‘Raio de Sol’ and ‘Manteiga’, all inoculation treatments produced the highest shoot (Figure 9b) and root (Figure 9c) biomass. In ‘Raio de Sol’, the three inoculation treatments did not differ significantly from each other and increased shoot and root dry mass by 68% and 78%, respectively, compared with the control. In ‘Manteiga’, *B. elkanii*, *A. brasilense*, and co-inoculation increased shoot dry mass by 166%, 343%, and 348%, respectively, while root dry mass rose by an average of 527% across inoculated treatments.

In contrast, ‘Orelha-de-vó’ exhibited the highest shoot and root biomass in control plants, indicating a negative response to inoculation. In ‘Cearense’, the control, *A. brasilense*, and co-inoculated plants showed greater biomass accumulation in both organs, whereas *B. elkanii* alone produced the lowest shoot and root dry masses.

### 2.5. Soil Fertility Indicators

In addition to plant traits, soil fertility parameters were evaluated after the cultivation of lima bean varieties inoculated with *B. elkanii*, *A. brasilense*, their co-inoculation, and the non-inoculated control. Factorial ANOVA revealed significant treatment effects on soil pH, electrical conductivity of the saturated paste extract (EC_e_), and the concentrations of phosphorus, potassium, calcium, and magnesium.

In soils cultivated with the variety ‘Raio de Sol’, co-inoculation with *B. elkanii* and *A. brasilense* increased pH to 7.42 compared with 6.7 in the non-inoculated control (Figure 10a). Conversely, in ‘Cearense’, inoculation with *A. brasilense* and co-inoculation resulted in a mean pH of 6.7, while the control soil reached 7.4. No significant pH differences were detected among treatments for the remaining varieties.

For soils cultivated with ‘Raio de Sol’, ‘Manteiga’, and ‘Cearense’, the highest EC_e_ values were observed in the control, whereas single and combined inoculations markedly reduced EC_e_ (Figure 10b). In ‘Raio de Sol’, inoculation with *A. brasilense* and co-inoculation reduced EC_e_ by an average of 45%. In ‘Manteiga’, all inoculation treatments decreased EC_e_ by approximately 55%, and in ‘Cearense’ by about 40% relative to the control. In contrast, in ‘Orelha-de-vó’, higher EC_e_ values were recorded in soils from plants inoculated with *A. brasilense* or co-inoculated.

In soils cultivated with ‘Raio de Sol’, inoculation with *B. elkanii* increased phosphorus concentration by 67% compared with the control (Figure 11a). Similarly, in ‘Orelha-de-vó’, both single and combined inoculations raised soil P by an average of 81% relative to the control. For the other varieties, phosphorus (Figure 11a) and potassium (Figure 11b) concentrations did not differ significantly among treatments. Regarding potassium, inoculation effects were particularly notable: in ‘Raio de Sol’, *B. elkanii* inoculation increased soil K by 50% relative to the control, and in ‘Orelha-de-vó’, co-inoculation enhanced soil K by 48%.

For soil calcium (Figure 11c), no significant differences were detected between control and inoculated treatments in ‘Raio de Sol’ and ‘Orelha-de-vó’. In ‘Manteiga’, inoculation with *A. brasilense* increased soil Ca by 53% compared with the control. Conversely, in ‘Cearense’, the control exhibited the highest soil Ca, with all inoculated treatments averaging 44% lower.

For soil magnesium (Figure 11d), no significant differences were observed among treatments in ‘Raio de Sol’, ‘Manteiga’, and ‘Orelha-de-vó’. However, in ‘Cearense’, soil Mg was highest in the control and decreased by approximately 47% under *A. brasilense* inoculation and co-inoculation.

## 3. Discussion

### 3.1. Nodulation and Nitrogen Metabolism

Overall, inoculation with *B. elkanii* and *A. brasilense* strains, either individually or in combination, significantly enhanced nodulation and nitrogen accumulation in different compartments of *P. lunatus*, with responses varying by genotype (Figure 1 and Figure 3). These responses are not determined solely by the inoculated microorganisms, but also by intrinsic genotypic differences, including root architecture, inherent nodulation efficiency, and genetic patterns of assimilate allocation, which collectively influence the dynamics of nitrogen uptake and redistribution [41,42].

These differences in nitrogen distribution among the evaluated varieties indicate that genotype plays a decisive role in nodulation capacity and in the internal regulation of nitrogen acquisition and allocation, reflecting inherent physiological and metabolic traits. This response is consistent with previous studies reporting enhanced nitrogen uptake and productivity in legumes subjected to co-inoculation, particularly under conditions of limited mineral nitrogen availability [32,33,40,43]. Moreover, the literature demonstrates that contrasting legume genotypes can differ markedly in nodulation efficiency, rates of symbiotic nitrogen fixation, and patterns of transport and utilization of reduced nitrogen, even when inoculated with the same microbial strains [44,45,46].

The inoculation treatments induced distinct redistributions of nitrogen between nodules, roots and shoots, following genotype-specific patterns. The varieties ‘Raio de Sol’ and ‘Cearense’ were associated with an increase in the number of nodules (Figure 1) and a higher nodular N content (Figure 3c), indicating an increase in N transfer to the air tissues via nodular metabolism. The simultaneous increase in N levels in nodules and shoots suggests efficient translocation of reduced N and its incorporation into leaf reserves, while the high N concentrations observed in the shoot may reflect genetic differences in xylem transport efficiency and leaf assimilation rates [47,48,49,50].

Root responses also varied among genotypes. In ‘Cearense’, *B. elkanii* inoculation in-creased root N content (Figure 3b), likely reflecting enhanced inorganic N uptake or retention [51]. Conversely, in ‘Manteiga’, *A. brasilense* and co-inoculation treatments reduced root N relative to the control, possibly due to greater N remobilization toward shoots or shifts in N partitioning. This response is also consistent with genotypes that prioritize N allocation to aerial tissues, a characteristic reported in other legumes with higher photosynthetic demand [50,52].

The data on nodules reinforces the central role of inoculation in modulating nitrogen metabolism. In all inoculated varieties, nodular N content was significantly higher than in the control (Figure 3c). Combined with the pink coloration observed in the nodules, this finding indicates elevated metabolic activity, reflecting increased leghemoglobin synthesis and nitrogenase activity that facilitate the export of reduced N to plant tissues [53,54]. The observed nodular and foliar N accumulation points to an improvement in both symbiotic fixation and systemic nitrogen assimilation.

The low nodular N content observed in the ‘Raio de Sol’ variety, together with the absence of nodules in non-inoculated plants under mineral N fertilization (control) (Figure 3c), is likely associated with mineral nitrogen–induced inhibition of nodule organogenesis and functionality. This inhibitory effect is mediated by autoregulatory mechanisms and nitrate-dependent signaling pathways that suppress the early stages of nodulation and symbiotic establishment [47,55,56,57]. Such regulatory processes plausibly explain the contrasting responses observed between fertilized and inoculated plants. In this context, mineral nitrogen acts on the plant regulatory system by activating nitrate signaling pathways that repress key nodulation-related genes, including *NIN*, *ENOD40*, and other regulators of the symbiotic signaling cascade [58].

The increases in foliar NH_4_^+^ (Figure 4a) and NO_3_^−^ (Figure 4b) pools observed in inoculated treatments provide additional insight into nitrogen metabolism after uptake, suggesting enhanced symbiotic N input, greater root absorption, and/or altered assimilation dynamics [40,56,57]. These results indicate that inoculation and co-inoculation affect both nitrogen acquisition and post-uptake processing, though distinguishing the relative contributions of BNF, assimilation, and redistribution would require isotopic or enzymatic approaches.

In summary, most inoculated plants exhibited concurrent increases in N content across nodules, roots, and leaves, demonstrating improved integration of nitrogen fixation, uptake, and redistribution. Elevated nodular N reflects intensified symbiotic activity, while higher root and foliar N denote efficient translocation of assimilated nitrogen to the shoots. Although confirmation via ^15^N tracing and enzymatic assays would refine this interpretation, the current evidence clearly supports the potential of inoculation to optimize nitrogen allocation in lima bean.

### 3.2. Plant Nutrition and Growth

Inoculation with *B. elkanii* and *A. brasilense*, individually or in combination, induced complex adjustments in the uptake and redistribution of mineral nutrients, reflected in variations in P, K, Ca, and Mg contents across *P. lunatus* organs (Figure 5, Figure 6, Figure 7 and Figure 8). These effects were strongly genotype-dependent, indicating that host-microbe compatibility influences nutrient transport and metabolic regulation [59,60,61].

The nutritional improvements were particularly evident in the increased P and K contents in shoots (Figure 5a and Figure 6a) in specific varieties and inoculation combinations, indicating enhanced nutrient uptake and preferential allocation to aerial tissues. The increase in shoot P content is consistent with previous reports showing that *Bradyrhizobium* and *Azospirillum* species can contribute to improved P acquisition through mechanisms such as phosphate solubilization or stimulation of root uptake processes [62,63,64]. Although physiological parameters related to P metabolism were not directly assessed in this study, the greater P accumulation observed in inoculated plants suggests a potential contribution to metabolic processes associated with growth, as reported for other legume–PGPR systems [65,66,67].

Similarly, increased K accumulation in shoots (Figure 6a) indicates improved potassium uptake and translocation to aerial tissues [68]. Potassium is widely recognized for its role in osmotic regulation, enzyme activation, and overall plant physiological performance [69,70], and its greater accumulation in shoots may be associated with the enhanced vegetative growth observed in specific varieties (Figure 9). Although root morphological traits and microbial solubilization processes were not directly evaluated, the observed increases in K content are consistent with previous studies reporting that microbial inoculation can favor nutrient acquisition through indirect effects on root functioning or rhizosphere processes [71,72,73,74]. In addition, higher P and K concentrations detected in roots under certain inoculation treatments (Figure 5b and Figure 6b) suggest improved nutrient absorption and retention capacity, reinforcing the contribution of microbial inoculation to plant mineral nutrition.

Changes in Ca and Mg contents (Figure 7 and Figure 8) further support the hypothesis of inoculant-induced nutritional modulation. In the ‘Orelha-de-vó’ variety, a 50% increase in shoot Ca under *A. brasilense* inoculation suggests a genotype-dependent enhancement of Ca^2+^ transport to aerial tissues. The 48% increase in root Mg in the same variety following *A. brasilense* inoculation is particularly relevant, since magnesium acts as a cofactor for enzymes involved in nitrogen assimilation and photosynthesis; this increase may indicate improved integration of carbon-nitrogen metabolism mediated by microbial activity [75,76,77].

In nodules, the substantial accumulation of macronutrients (Figure 5c, Figure 6c, Figure 7c and Figure 8c) confirms that inoculation not only enhances nodule formation (Figure 1) but also is associated with higher nutrient contents within these structures, as previously reported [47,78,79]. Higher P content in nodules is commonly associated in the literature with ATP availability required to sustain nitrogenase function, whereas enrichment in K and Ca has been linked to ionic homeostasis and structural stability of nodules, which are considered important for symbiotic efficiency [54,80,81].

These nutritional changes, coupled with the effects on nitrogen metabolism de-scribed in Section 3.1, directly influenced plant growth and biomass accumulation (Figure 9). The ‘Raio de Sol’ and ‘Manteiga’ varieties exhibited marked increases in plant height (Figure 9a) and shoot (Figure 9b) and root dry masses (Figure 9c), reflecting improved nutrient acquisition and redistribution promoted by microbial activity. In the ‘Manteiga’ variety, this response was particularly pronounced, suggesting a higher intrinsic responsiveness to microbial inoculation, possibly associated with genotype-specific traits related to root architecture, assimilate allocation, and nitrogen use efficiency. The positive association between N, P, and K contents and biomass reinforces the role of these macronutrients in regulating vegetative growth in the evaluated plants.

Conversely, the ‘Orelha-de-vó’ and ‘Cearense’ varieties displayed distinct and sometimes inverse responses, suggesting genotype-dependent differences in symbiotic compatibility and nutrient uptake efficiency. These variations indicate that inoculation efficiency depends on host genotype. In some genotypes, resource allocation has been reported in the literature to favor reproductive rather than vegetative development [82,83,84], which may partly explain the observed response patterns. Further studies including yield evaluation are necessary to verify these tendencies in lima bean.

Overall, the results demonstrate that inoculation with *B. elkanii* and *A. brasilense* significantly influences mineral nutrition in *P. lunatus*, enhancing macronutrient uptake and redistribution and promoting greater biomass accumulation. This response reflects the integration of increased root absorption, intensified nodular metabolism, and improved nutrient translocation within the plant. Nevertheless, the similarity between co-inoculation treatments and single inoculations suggests that, although effective, the combined effects were not strictly additive. This outcome may be explained by functional redundancy between *B. elkanii* and *A. brasilense*, such as shared roles in nutrient solubilization or stimulation of root growth, or by interspecific interactions, including competition for resources in the rhizosphere or cross-regulation of plant signaling pathways [85]. For example, phytohormones produced by *A. brasilense* may influence nodulation signaling mediated by *Bradyrhizobium*, as bacterial auxins such as indole-3-acetic acid have been shown to affect root development and the establishment of rhizobial symbiosis in legumes [86].

### 3.3. Changes in Soil Fertility Indicators

The baseline soil conditions provide essential context for interpreting the observed effects. Before cultivation, the soil exhibited a pH of 5.8, an EC_e_ of 0.07 dS m^−1^, and a cation exchange capacity (CEC) of 5.04 cmol_c_ dm^−3^, characterizing it as moderately acidic, low in salinity, and with limited cation exchange capacity. These properties indicate sensitivity to ionic fluctuations and microbial-induced changes throughout the experiment.

The increase in pH observed after cultivation was consistent across several treatments, with values rising from 5.8 to near or above 7.0 in certain cases (Figure 10a). In the ‘Raio de Sol’ variety, co-inoculation elevated pH to 7.42, whereas in ‘Cearense’, *A. brasilense* inoculation and co-inoculation produced mean pH values lower than those of the control. These results demonstrate genotype-specific and treatment-dependent shifts, consistent with evidence that rhizosphere microorganisms can modulate local H^+^/OH^−^ balance by altering proton fluxes, organic acid secretion, and root exudation patterns, thereby influencing rhizosphere pH and ion uptake [87,88,89].

The initially very low EC_e_ indicates that the increases observed in the soils of the control treatments across all varieties (Figure 10b) were primarily attributable to salt inputs, likely associated with the irrigation water used during the experiment, which had an electrical conductivity (EC_w_) of 0.45 dS m^−1^. Despite the low EC_w_, the irrigation water still contributed salts to the soil. In contrast, several inoculation treatments reduced soil EC_e_ relative to the control in the ‘Raio de Sol’, ‘Manteiga’, and ‘Cearense’ varieties. These reductions suggest that inoculation modulated the dynamics of soluble salts in the soil, promoting lower ionic accumulation and decreased electrical conductivity, even under initially very low salinity conditions, as reported in other studies [90,91,92].

Soil P contents (Figure 11a) exhibited distinct behaviors among treatments and varieties. In control soils, values remained close to the pre-cultivation level (15.13 mg dm^−3^), whereas certain treatments increased P availability, particularly inoculation with *B. elkanii* in the ‘Raio de Sol’ variety and all three inoculation forms in the ‘Orelha-de-vó’ variety. These increases indicate alterations in phosphorus availability within the soil–plant system associated with inoculation [65], although the specific mechanisms cannot be determined without complementary data on mineralization, sorption, or enzymatic activity.

For soil potassium (Figure 11b), the control showed a decrease relative to the initial value of 116.42 mg dm^−3^, whereas specific inoculations promoted relative increases in available K. Increases were observed in ‘Raio de Sol’ with *B. elkanii* and in ‘Orelha-de-vó’ with co-inoculation. These results suggest potential microbial mobilization or solubilization of K in the soil, consistent with reports that microorganisms can release K from poorly available minerals and increase the soluble potassium fraction in soil–plant systems [93,94].

Soil Ca contents (Figure 11c) increased relative to the initial level (1.40 cmol_c_ dm^−3^) in all treatments, including the control. However, responses to inoculation varied among varieties. The increases observed in ‘Manteiga’ and ‘Orelha-de-vó’ indicate a positive effect of specific inoculations on soil Ca availability, whereas the reductions in Ca (Figure 11c) and Mg (Figure 11d) recorded in ‘Cearense’ under inoculation suggest variety-specific changes in the dynamics of these ions. Further studies are needed to elucidate the mechanisms underlying these responses.

The contribution of irrigation water as a source of salts likely explains the increases in EC_e_, Ca, and Mg observed in the control. The low CEC and moderate organic matter content of the initial soil (10.66 g kg^−1^) may have amplified these ionic variations, given the limited buffering capacity of the system. Accordingly, the variations observed in P and K under inoculation can be interpreted in light of this baseline, which favors rapid adjustments in the soluble nutrient fraction.

Overall, microbial inoculation modified soil fertility indicators in a strain- and *P. lunatus* variety-dependent manner, starting from a moderately acidic soil with low salinity and limited buffering capacity. The integration of soil and plant variables revealed correlations between nutrient availability and plant nutritional parameters. Treatments that increased soil P and K availability generally coincided with higher nutrient contents in shoots, whereas reductions in EC_e_ under inoculation occurred in treatments that also exhibited improved plant performance in certain varieties. These associations suggest that inoculants influenced nutrient dynamics within the soil–plant system, although the specific mechanisms cannot be fully elucidated with the current data.

## 4. Materials and Methods

### 4.1. Location and Experimental Conditions

The experiment was conducted in a greenhouse at the Center for Agricultural and Environmental Sciences, Paraíba State University (UEPB), located in Lagoa Seca, Paraíba State, Brazil. Plants were cultivated in polyethylene pots containing 5.0 kg of soil collected from the experimental field of UEPB, in Lagoa Seca, PB, with medium texture, which presented the following chemical characteristics prior to the experiment: pH = 5.8; ECe = 0.07 dS m^−1^; P, K^+^, and Na^+^ = 15.13, 116.42, and 0.13 mg dm^−3^, respectively; H^+^ + Al^3+^, Al^3+^, Ca^2+^, Mg^2+^, and CEC = 2.56, 0.15, 1.40, 0.65, and 5.04 cmol_c_ dm^−3^; and organic matter = 10.66 g kg^−1^.

Soil was collected from a 0–20 cm depth, sieved to remove clods, roots, and other residues, and placed in pots. Before sowing, soil moisture was adjusted to field capacity, and plants were irrigated manually on a daily basis using water from a reservoir (pH = 8.0, EC_w_ = 0.45 dS m^−1^). The irrigation volume was adjusted daily according to plant water requirements, determined through a drainage lysimetry system.

### 4.2. Treatments and Plant Material

Four lima bean (*P. lunatus*) varieties (‘Raio de Sol’, ‘Manteiga’, ‘Orelha-de-vó’, and ‘Cearense’) were evaluated under four cultivation conditions: non-inoculated control, inoculation with *B. elkanii*, inoculation with *A. brasilense*, and co-inoculation with *B. elkanii* and *A. brasilense*. The experiment followed a completely randomized design in a 4 × 4 factorial arrangement with four replicates, totaling 64 experimental units. Each experimental unit consisted of one plant.

In the control treatment, nitrogen fertilization was performed 10 days after sowing (DAS), following crop recommendations, with the application of 63.6 mg N kg^−1^ of soil using ammonium sulfate as the nitrogen source, without inoculants. The seeds of the studied varieties are shown in Figure 12, highlighting their distinguishing characteristics.

### 4.3. Preparation of Inoculants

The inoculants were prepared using the bacterial strains *B. elkanii* BR 2003 and *A. brasilense* Ab-V5, both isolated and provided by the Brazilian Agricultural Research Corporation (Embrapa Agrobiologia, Rio de Janeiro, Brazil). Initially, the bacteria were cultured on YM agar plates (1% glucose, 2% agar, 0.5% peptone, 0.3% malt extract, and 0.3% yeast extract) for five days at 28 °C in a BOD-type growth chamber. Subsequently, the microorganisms were transferred to liquid Yeast Extract Malt (YEM) medium, with the same composition as the solid medium, and incubated at 28 °C under constant agitation (150 rpm) for seven days, until reaching the late exponential growth phase (1.0 × 10^9^ CFU mL^−1^). The end of the growth phase was confirmed by a color change in the medium caused by bacterial metabolic activity. After preparation, the inoculants were ready for seed application.

### 4.4. Seed Disinfection, Inoculation, and Sowing

Before sowing, lima bean seeds were surface-sterilized by immersion in pure ethanol for 30 s, followed by immersion in a 1% sodium hypochlorite solution for three minutes, and rinsed ten times with sterile distilled water. Inoculation consisted of applying 1 mL of inoculant per seed directly onto the seed surface using a micropipette at sowing, followed by covering with soil. Four seeds were sown per pot, and ten days after germination, thinning was performed to maintain one plant per pot.

### 4.5. Experimental Analyses

At 80 DAS, plants were evaluated for root nodule formation, nitrogen accumulation in different organs, macronutrient accumulation, growth, and biomass production. On the same day, soil samples were collected from each pot for chemical analysis.

#### 4.5.1. Root Nodule Formation

Roots were carefully removed from the pots, washed, and nodules manually detached. Samples were dried in a forced-air oven at 65 °C until constant weight and weighed on a precision analytical balance to determine dry mass. The total number of root nodules and their dry mass were recorded.

#### 4.5.2. Nitrogen Accumulation Analysis

Nitrogen (N) accumulation in plant organs was determined using the Kjeldahl method, following Tedesco et al. [95]. Plants were separated into shoots, roots, and nodules, dried at 65 °C until constant weight, and ground in a Wiley mill.

For distillation, 20 mL of the digested extract were transferred to a digestion tube connected to the distillation apparatus. In a 125 mL Erlenmeyer flask, 10 mL of boric acid solution containing indicator were added, and 10 mL of 13 N NaOH were placed in the distiller inlet cup. The distiller valve was opened gradually to mix the alkaline solution with the tube contents. Distillation was performed, and the released ammonia was absorbed in the boric acid solution until reaching approximately 50 mL. The absorbed ammonia was titrated with 0.07143 N HCl, and the endpoint was determined by a color change from green to dark pink.

Nitrogen content was calculated based on the volume of HCl consumed, determining N concentration in shoots, roots, and nodules. Additionally, ammonium (NH_4_^+^) and nitrate (NO_3_^−^) contents in shoots were determined according to Thomas et al. [96] and Bremner et al. [97].

#### 4.5.3. Accumulation of Other Nutrients in Different Plant Organs

Phosphorus, potassium, calcium, and magnesium were determined in shoot, root, and nodules.

Phosphorus was analyzed according to Murphy and Riley [98], potassium was measured by flame photometry (Tecnal, model 910, Piracicaba, Brazil) [99], and calcium and magnesium were quantified by atomic absorption spectrometry (GBC Scientific Equipment, model SavantAA, Melbourne, Australia) [100].

#### 4.5.4. Plant Growth and Biomass Analysis

Plant growth was assessed by measuring the main stem length (plant height) from the collar to the apex using a millimeter-graduated measuring tape. Shoot and root dry masses were determined after drying in a forced-air oven at 65 °C until constant weight was achieved.

#### 4.5.5. Soil Chemical Properties After Plant Cultivation

Soil pH, electrical conductivity of the saturated soil extract (EC_e_), and the contents of P, K, Ca, and Mg were evaluated. The quantification of P, K, Ca, and Mg followed the procedures described in the *Manual of Chemical Analysis of Soils*, *Plants*, *and Fertilizers* [99].

Soil pH (H_2_O) was determined using a bench-top digital microprocessor pH meter (MS Tecnopon, model DL-PH, Piracicaba, Brazil), according to Donagema et al. [101]. EC_e_ was measured with a conductivity meter (MS Tecnopon, model MCA-150, Piracicaba, Brazil), following the methodology described by Silva [99]. For both determinations, 10 g of soil were mixed with 25 mL of deionized water, stirred with a glass rod, and allowed to stabilize for 30 min. The suspension was then homogenized, after which pH was measured using a pH meter, and EC_e_ was determined with a bench-top conductivity meter.

### 4.6. Statistical Analysis

Data normality was verified using the Shapiro–Wilk test, and homogeneity of variances was assessed with Bartlett’s test. When assumptions were satisfied, data were analyzed by factorial ANOVA (*p* ≤ 0.05). Mean comparisons were performed using Tukey’s HSD test (*p* ≤ 0.05).

## 5. Conclusions

The results of this study demonstrate that inoculation of *P. lunatus* with *B. elkanii* (strain BR 2003) and *A. brasilense* (strain Ab-V5), applied either individually or in combination, enhances nodulation, nitrogen assimilation, and mineral nutrition, with genotype-dependent responses. Inoculated plants consistently showed improved symbiotic performance and more efficient redistribution of nitrogen among plant organs, resulting in greater nutritional status and vegetative development, particularly in the most responsive varieties.

The observed increases in growth, biomass accumulation, and nutrient uptake support the first hypothesis of this study, indicating that microbial inoculation is an effective strategy to improve the physiological performance of lima bean plants. Importantly, co-inoculation produced responses comparable to those obtained with single inoculations, indicating that the benefits were not additive but consistent across inoculation strategies.

Regarding the second hypothesis, microbial inoculation also influenced post-cultivation soil chemical attributes in specific genotype–microorganism combinations. Improvements in soil pH, availability of phosphorus and potassium, and reductions in electrical conductivity suggest that inoculation can contribute positively to residual soil fertility, although these effects were genotype-dependent and not uniformly observed across all treatments.

Overall, the findings indicate that inoculation with *B. elkanii* and *A. brasilense*, whether applied individually or in combination, represents a viable alternative to conventional nitrogen fertilization in lima bean cultivation. The similarity of responses among inoculation strategies highlights the importance of genotype–microbe compatibility rather than the inoculation approach itself. Future studies should focus on long-term field evaluations and on identifying the most responsive genotype–inoculant combinations to support sustainable management practices in semiarid environments.

## Figures and Tables

**Figure 1 plants-15-00135-f001:**
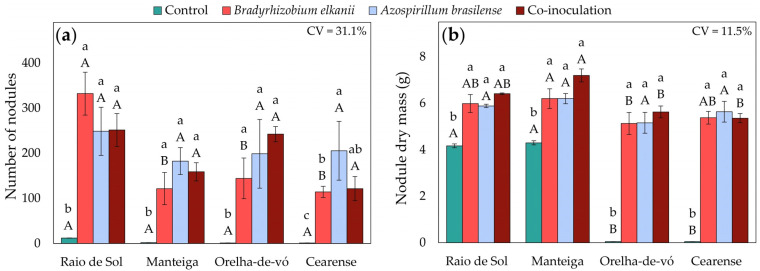
Number of nodules (**a**) and nodule dry mass (**b**) of four *Phaseolus lunatus* varieties subjected to the following treatments: control (fertilization with nitrogen), inoculation with *Bradyrhizobium elkanii*, inoculation with *Azospirillum brasilense*, and co-inoculation (*B. elkanii* + *A. brasilense*). Different lowercase letters indicate significant differences among inoculation treatments within each variety, and different uppercase letters indicate significant differences among varieties within each inoculation, according to Tukey’s HSD test (*p* ≤ 0.05). Bars represent the standard error of the mean (*n* = 4). CV: coefficient of variation.

**Figure 2 plants-15-00135-f002:**
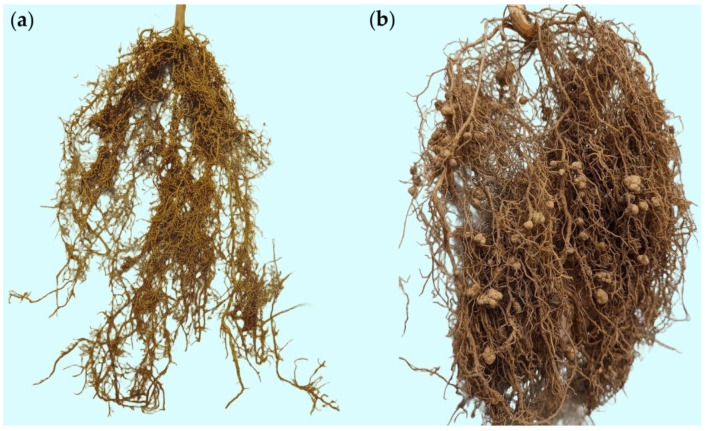
Representative photographs of *Phaseolus lunatus* (‘Raio de Sol’ variety) roots at 80 days after sowing (DAS) under non-inoculated control (**a**) and inoculation with *Bradyrhizobium elkanii* (**b**).

**Figure 3 plants-15-00135-f003:**
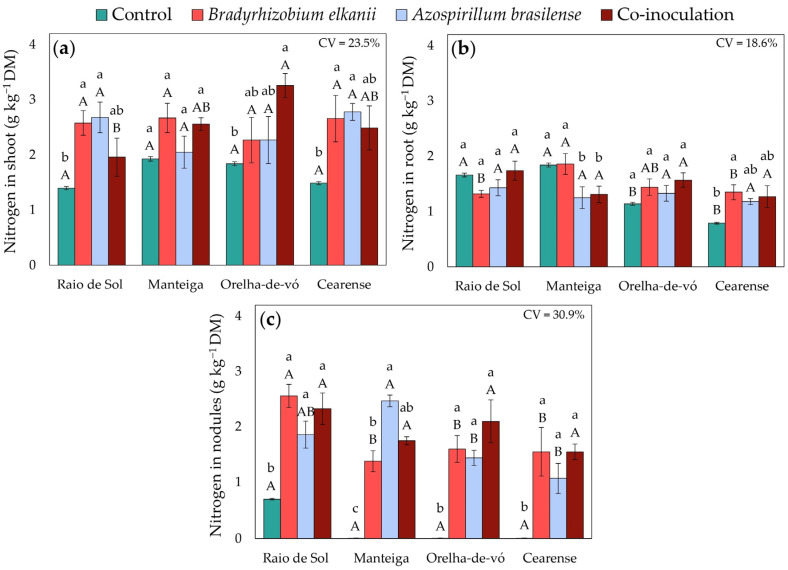
Nitrogen contents in shoots (**a**), roots (**b**), and nodules (**c**) of four *Phaseolus lunatus* varieties subjected to the following treatments: control (fertilization with nitrogen), inoculation with *Bradyrhizobium elkanii*, inoculation with *Azospirillum brasilense*, and co-inoculation (*B. elkanii* + *A. brasilense*). Different lowercase letters indicate significant differences among inoculation treatments within each variety, and different uppercase letters indicate significant differences among varieties within each inoculation, according to Tukey’s HSD test (*p* ≤ 0.05). Bars represent the standard error of the mean (*n* = 4). CV: coefficient of variation.

**Figure 4 plants-15-00135-f004:**
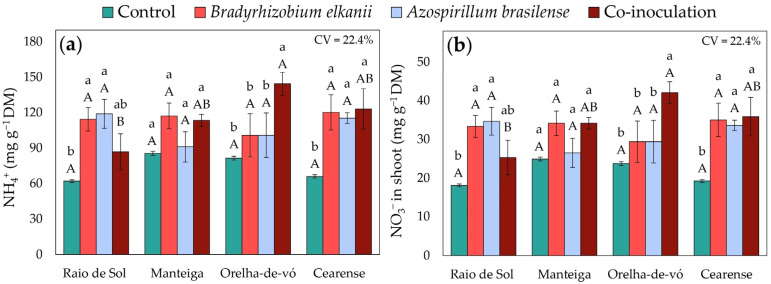
NH_4_^+^ (**a**) and NO_3_^−^ (**b**) in shoots of four *Phaseolus lunatus* varieties subjected to the following treatments: control (fertilization with nitrogen), inoculation with *Bradyrhizobium elkanii*, inoculation with *Azospirillum brasilense*, and co-inoculation (*B. elkanii* + *A. brasilense*). Different lowercase letters indicate significant differences among inoculation treatments within each variety, and different uppercase letters indicate significant differences among varieties within each inoculation, according to Tukey’s HSD test (*p* ≤ 0.05). Bars represent the standard error of the mean (*n* = 4). CV: coefficient of variation.

**Figure 5 plants-15-00135-f005:**
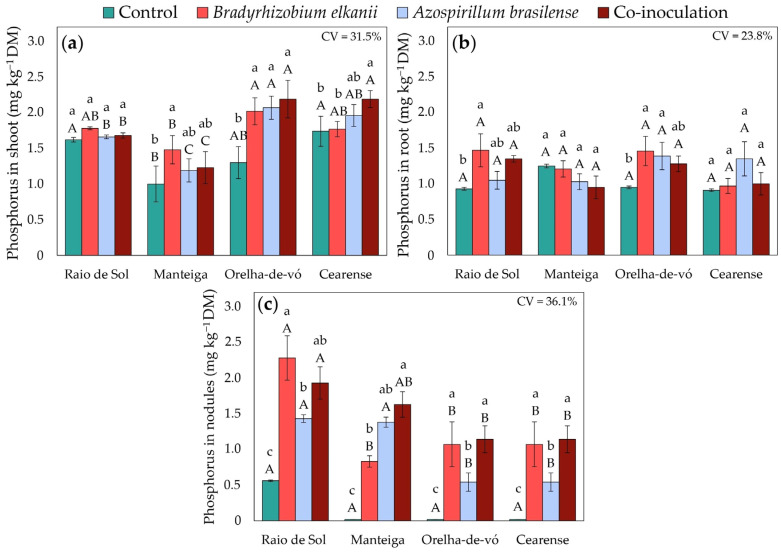
Phosphorus contents in shoots (**a**), roots (**b**), and nodules (**c**) of four *Phaseolus lunatus* varieties subjected to the following treatments: control (fertilization with nitrogen), inoculation with *Bradyrhizobium elkanii*, inoculation with *Azospirillum brasilense*, and co-inoculation (*B. elkanii* + *A. brasilense*). Different lowercase letters indicate significant differences among inoculation treatments within each variety, and different uppercase letters indicate significant differences among varieties within each inoculation, according to Tukey’s HSD test (*p* ≤ 0.05). Bars represent the standard error of the mean (*n* = 4). CV: coefficient of variation.

**Figure 6 plants-15-00135-f006:**
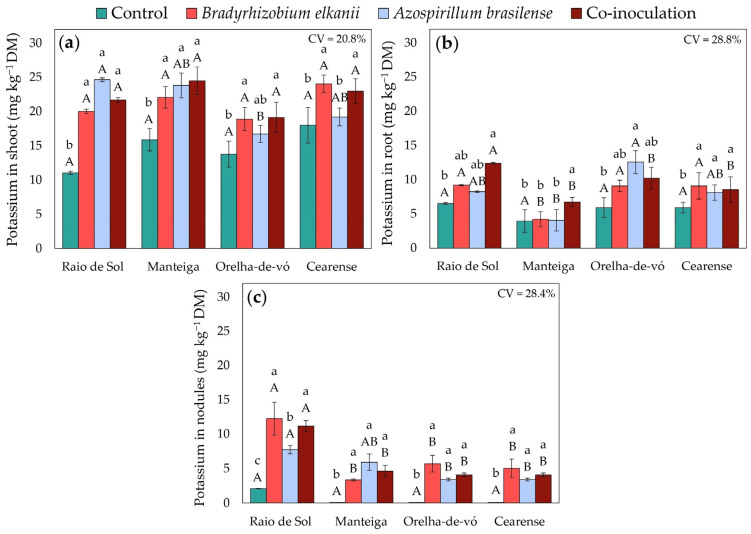
Potassium contents in shoots (**a**), roots (**b**), and nodules (**c**) of four *Phaseolus lunatus* varieties subjected to the following treatments: control (fertilization with nitrogen), inoculation with *Bradyrhizobium elkanii*, inoculation with *Azospirillum brasilense*, and co-inoculation (*B. elkanii* + *A. brasilense*). Different lowercase letters indicate significant differences among inoculation treatments within each variety, and different uppercase letters indicate significant differences among varieties within each inoculation, according to Tukey’s HSD test (*p* ≤ 0.05). Bars represent the standard error of the mean (*n* = 4). CV: coefficient of variation.

**Figure 7 plants-15-00135-f007:**
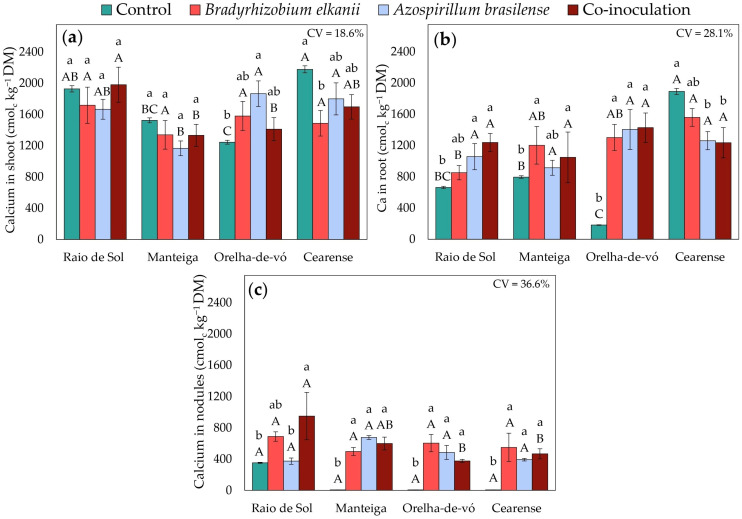
Calcium contents in shoots (**a**), roots (**b**), and nodules (**c**) of four *Phaseolus lunatus* varieties subjected to the following treatments: control (fertilization with nitrogen), inoculation with *Bradyrhizobium elkanii*, inoculation with *Azospirillum brasilense*, and co-inoculation (*B. elkanii* + *A. brasilense*). Different lowercase letters indicate significant differences among inoculation treatments within each variety, and different uppercase letters indicate significant differences among varieties within each inoculation, according to Tukey’s HSD test (*p* ≤ 0.05). Bars represent the standard error of the mean (*n* = 4). CV: coefficient of variation.

**Figure 8 plants-15-00135-f008:**
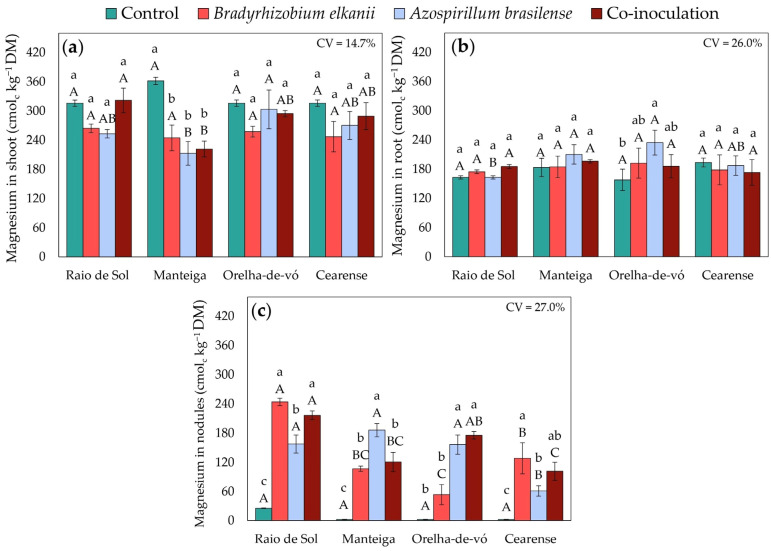
Magnesium contents in shoots (**a**), roots (**b**), and nodules (**c**) of four *Phaseolus lunatus* varieties subjected to the following treatments: control (fertilization with nitrogen), inoculation with *Bradyrhizobium elkanii*, inoculation with *Azospirillum brasilense*, and co-inoculation (*B. elkanii* + *A. brasilense*). Different lowercase letters indicate significant differences among inoculation treatments within each variety, and different uppercase letters indicate significant differences among varieties within each inoculation, according to Tukey’s HSD test (*p* ≤ 0.05). Bars represent the standard error of the mean (*n* = 4). CV: coefficient of variation.

**Figure 9 plants-15-00135-f009:**
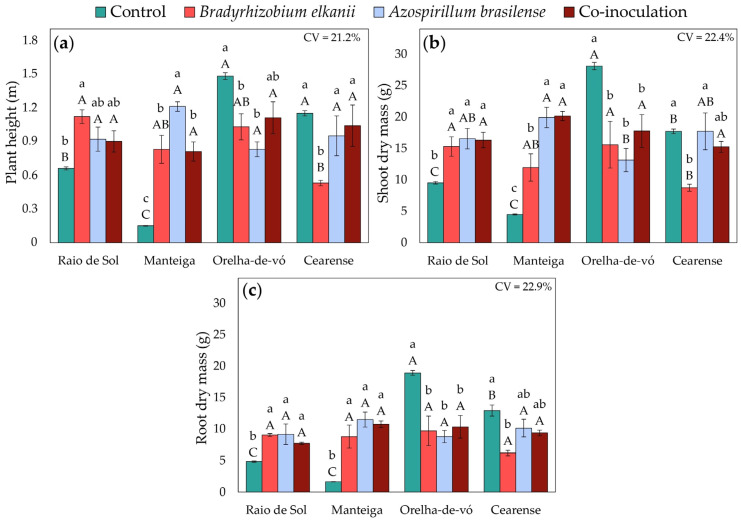
Plant height (**a**), shoot dry mass (**b**), and root dry mass (**c**) in four *Phaseolus lunatus* varieties subjected to the following treatments: control (fertilization with nitrogen), inoculation with *Bradyrhizobium elkanii*, inoculation with *Azospirillum brasilense*, and co-inoculation (*B. elkanii* + *A. brasilense*). Different lowercase letters indicate significant differences among inoculation treatments within each variety, and different uppercase letters indicate significant differences among varieties within each inoculation, according to Tukey’s HSD test (*p* ≤ 0.05). Bars represent the standard error of the mean (*n* = 4). CV: coefficient of variation.

**Figure 10 plants-15-00135-f010:**
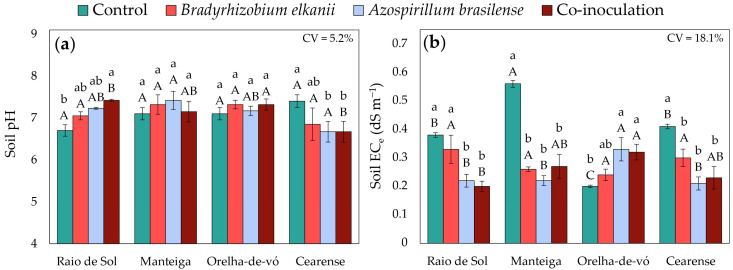
Soil pH (**a**) and electrical conductivity of the saturated paste extract (EC_e_) (**b**) in soil after cultivation with four *Phaseolus lunatus* varieties subjected to the following treatments: control (fertilization with nitrogen), inoculation with *Bradyrhizobium elkanii*, inoculation with *Azospirillum brasilense*, and co-inoculation (*B. elkanii* + *A. brasilense*). Different lowercase letters indicate significant differences among inoculation treatments within each variety, and different uppercase letters indicate significant differences among varieties within each inoculation, according to Tukey’s HSD test (*p* ≤ 0.05). Bars represent the standard error of the mean (*n* = 4). CV: coefficient of variation.

**Figure 11 plants-15-00135-f011:**
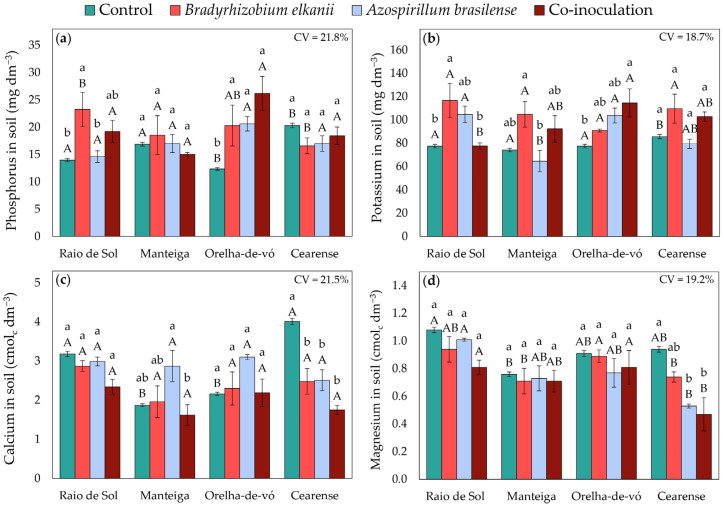
Phosphorus (**a**), potassium (**b**), calcium (**c**), and magnesium (**d**) contents in soil after cultivation with four *Phaseolus lunatus* varieties subjected to the following treatments: control (fertilization with nitrogen), inoculation with *Bradyrhizobium elkanii*, inoculation with *Azospirillum brasilense*, and co-inoculation (*B. elkanii* + *A. brasilense*). Different lowercase letters indicate significant differences among inoculation treatments within each variety, and different uppercase letters indicate significant differences among varieties within each inoculation, according to Tukey’s HSD test (*p* ≤ 0.05). Bars represent the standard error of the mean (*n* = 4). CV: coefficient of variation.

**Figure 12 plants-15-00135-f012:**
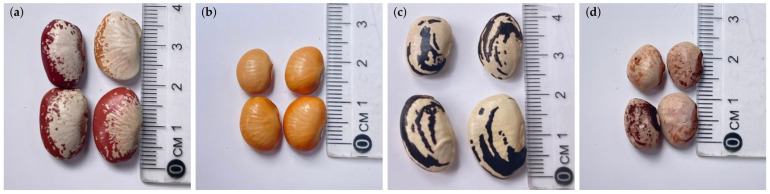
Seeds of the four lima bean (*Phaseolus lunatus*) varieties used in the experiment, showing distinguishing features: ‘Raio de Sol’ (**a**), ‘Manteiga’ (**b**), ‘Orelha-de-vó’ (**c**), and ‘Cearense’ (**d**).

## Data Availability

The dataset is available upon request from the authors. The data are not publicly available due to privacy restrictions.

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
