# Peer review of "Impact of *Bradyrhizobium elkanii* and *Azospirillum brasilense* Co-Inoculation on Nitrogen Metabolism, Nutrient Uptake, and Soil Fertility Indicators in *Phaseolus lunatus* Genotypes"

_plants, 2026, doi:10.3390/plants15010135_

Round 1

Reviewer 1 Report

Comments and Suggestions for Authors
  1. Abstract: The core advantage of co-inoculation should be clearly stated. Currently, it only mentions "inoculation strategies significantly increased...", without distinguishing the differences between single and co-inoculation (e.g., whether co-inoculation is superior in metrics like "nitrogen assimilation + soil pH improvement").

  2. The results in the abstract should be re-described by incorporating specific key data.

  3. Introduction: The innovation of this study needs further clarification. While it mentions "co-inoculation... remains poorly explored", a specific comparison can be made: what are the unique aspects of the interaction mechanisms between lima beans and these two bacterial strains compared to well-studied crops like soybean and common bean? Is the genetic background difference among the four tested varieties (e.g., 'Raio de Sol', 'Manteiga') clearly defined? Why were these specific varieties chosen? Characteristics of the varieties (e.g., locally dominant cultivars, differences in nodulation capacity) should be supplemented to support the research objective focusing on "genotype-dependent responses".

  4. Regarding the lack of a synergistic effect in co-inoculation (evidenced by overlapping clusters with single inoculation in the PCA plot), the microbial interaction mechanisms need discussion: Could interspecific competition exist (e.g., for nutrients, metabolite inhibition)? Or functional redundancy (e.g., both strains solubilizing phosphorus)? This could be explored by referencing literature on potential cross-regulation, such as between IAA production by Azospirillum and nodulation signaling pathways in Bradyrhizobium.

  5. The dramatic 707% increase in plant height for the 'Manteiga' variety under inoculation (Fig 5a), far exceeding other varieties, requires explanation from a varietal specificity perspective: Does this variety produce root exudates that more effectively attract PGPR? Or are there differences in the expression of genes related to nitrate reductase activity/transporters (e.g., NRT2)? Supplementing with physiological data (e.g., enzyme activity) or candidate gene expression data is recommended.

  6. Materials and Methods: The initial nitrogen content of the soil (e.g., total N, available N) should be provided to help explain whether the low nodule number in the "control (N-fertilized)" group results from "nitrogen inhibition effect" (as literature suggests high N can suppress rhizobial infection).

  7. The manuscript only mentions "1 mL inoculant per seed". The bacterial concentration (e.g., CFU/mL) and the inoculation timing (pre-sowing/seedling stage) should be supplemented to ensure experimental reproducibility.

  8. The terms "nodule dry mass" and "nodule dry weight" used in the manuscript should be unified.

  9. Some sentences in the Introduction and Discussion are structurally complex; simplifying them is advised to enhance readability.

Author Response

Reviewer's comment: Abstract: The core advantage of co-inoculation should be clearly stated. Currently, it only mentions "inoculation strategies significantly increased...", without distinguishing the differences between single and co-inoculation (e.g., whether co-inoculation is superior in metrics like "nitrogen assimilation + soil pH improvement").

Answer: Single inoculation also generates benefits, but co-inoculation provides the potentiation of the effects and reduces variability in the crop and experiment, especially under stress. Co-inoculation promotes increased microbial activity, greater nitrogen availability in the planting system, improves nutritional availability, and can attenuate the pH improvement with less acidification.

Reviewer's comment: Abstract: The results in the abstract should be re-described by incorporating specific key data.

Answer: The authors thank the reviewer for this suggestion. The Abstract was reformulated to incorporate specific key data, and the changes were implemented based on this comment as well as on suggestions provided by the other reviewers, ensuring consistency with the results presented in the main text.

Reviewer's comment: Introduction: The innovation of this study needs further clarification. While it mentions "co-inoculation... remains poorly explored", a specific comparison can be made: what are the unique aspects of the interaction mechanisms between lima beans and these two bacterial strains compared to well-studied crops like soybean and common bean? Is the genetic background difference among the four tested varieties (e.g., 'Raio de Sol', 'Manteiga') clearly defined? Why were these specific varieties chosen? Characteristics of the varieties (e.g., locally dominant cultivars, differences in nodulation capacity) should be supplemented to support the research objective focusing on "genotype-dependent responses".

Answer: Additional information about the innovation that the study brings about fava beans when compared to other crops inoculated with these strains and their differential were incorporated, different nodulation responses in each variety, as well as the points of use for the choice of each genotype and its adaptability to the Introduction, as recommended.

Reviewer's comment: Regarding the lack of a synergistic effect in co-inoculation (evidenced by overlapping clusters with single inoculation in the PCA plot), the microbial interaction mechanisms need discussion: Could interspecific competition exist (e.g., for nutrients, metabolite inhibition)? Or functional redundancy (e.g., both strains solubilizing phosphorus)? This could be explored by referencing literature on potential cross-regulation, such as between IAA production by Azospirillum and nodulation signaling pathways in Bradyrhizobium.

Answer: We have addressed this comment by adding a brief discussion in the Discussion section exploring possible mechanisms underlying the absence of a synergistic effect in the co-inoculation treatment. Specifically, we now discuss hypotheses related to functional redundancy between Bradyrhizobium elkanii and Azospirillum brasilense, potential interspecific competition in the rhizosphere, and cross-regulation of plant signaling pathways, including the possible influence of Azospirillum-derived IAA on nodulation signaling mediated by Bradyrhizobium, supported by relevant literature.

Reviewer's comment: The dramatic 707% increase in plant height for the 'Manteiga' variety under inoculation (Fig 5a), far exceeding other varieties, requires explanation from a varietal specificity perspective: Does this variety produce root exudates that more effectively attract PGPR? Or are there differences in the expression of genes related to nitrate reductase activity/transporters (e.g., NRT2)? Supplementing with physiological data (e.g., enzyme activity) or candidate gene expression data is recommended.

Answer: We thank the reviewer for this valuable comment. The additional physiological and molecular analyses suggested cannot be performed because no plant tissue remains available for enzyme activity or gene expression assays. Nevertheless, we have revised the Discussion to include a concise explanation indicating that the pronounced growth response of the ‘Manteiga’ variety may be related to varietal specificity, such as higher responsiveness to microbial inoculation associated with genotype-dependent traits in root system characteristics and nitrogen use efficiency. These interpretations are presented as literature-supported hypotheses and are clearly stated as not experimentally verified within the scope of this study.

Reviewer's comment: Materials and Methods: The initial nitrogen content of the soil (e.g., total N, available N) should be provided to help explain whether the low nodule number in the "control (N-fertilized)" group results from "nitrogen inhibition effect" (as literature suggests high N can suppress rhizobial infection).

Answer: The analysis of mineral nitrogen (available N) was not included due to the highly dynamic and transient nature of its ionic forms under tropical conditions, where rapid leaching and volatilization prevent a single initial sampling from accurately reflecting N availability during the root infection period. In Brazil, routine soil analyses do not usually include N because of this temporal instability, and the inhibition of nodulation observed in the control treatment in our study is therefore directly attributed to the controlled application of mineral fertilizer, applied at rates recommended based on soil analysis.

Reviewer's Comment: Materials and Methods

The manuscript only mentions "1 mL inoculant per seed". The bacterial concentration (e.g., CFU/mL) and the inoculation timing (pre-sowing/seedling stage) should be supplemented to ensure experimental reproducibility.

Answer: We appreciate the reviewer’s request for clarification regarding experimental reproducibility. We would like to clarify that the information regarding the bacterial concentration was already present in the manuscript (line 694 in the revised version). Furthermore, following your suggestion, the details concerning the inoculation timing (pre-sowing/seedling stage) have been explicitly added/supplemented in lines 701–702 to ensure the methodology is fully transparent.

Reviewer's comment: The terms "nodule dry mass" and "nodule dry weight" used in the manuscript should be unified.

Answer: The terms were unified throughout the manuscript.

Reviewer's comment: Some sentences in the Introduction and Discussion are structurally complex; simplifying them is advised to enhance readability.

Answer: The authors appreciate the important observation. In accordance with these recommendations, the following actions were taken in this new version of the manuscript: (i) new information was added, specifically regarding the state of the art of the topic in a global context, reinforcing the potential of the study for the development of the Brazilian semiarid region; (ii) some paragraphs were rewritten and organized to improve the quality of the text; (iii) detailed information was presented about the occurrence of different genotypes in different locations of the Brazilian semi-arid region; (iv) the research problem and justification were improved, and the study hypothesis was added; (v) some discussions were rewritten and/or new sentences were added to reduce the complexity of the text. The authors agree that the suggestions made contributed significantly to improving the quality of the manuscript.

Reviewer 2 Report

Comments and Suggestions for Authors

This study investigates the effects of single and combined inoculation with Bradyrhizobium elkanii (strain BR 2003) and Azospirillum brasilense (strain Ab-V5) on nitrogen metabolism, nutrient uptake, plant growth, and residual soil fertility in P. lunatus.

Although the topic is well exposed and aligns with the journal's scope, some parts of this manuscript need to be implemented (Introduction, PCA).

Major revisions are necessary to increase the overall quality of this article.

Abstract:

  • Lines 32-33. Add some numbers in this section for the main parameters investigated

(root nodulation, nitrogen assimilation rate, accumulation of P, K, Ca, and Mg in plant tissues) as a percentage of increase/decrease.

Introduction:

I find this section a little bit short and incomplete. Add more information about the importance of legumes in the human diet, their economic importance worldwide, main producers, etc., supported by the literature.  

Results:

  • Figure 3 and Figure 4: Indicate if the results are fresh-weight-based or dry-weight-based in the didascaly in order to facilitate the readers' understanding.

Discussions:

  • Lines 436-460. Multivariate Analysis of Plant Traits (PCA). The discussion of the PCA section is very short and incomplete, in my opinion. The authors should explain in detail the finding point by point and not give only some general argumentations.

Materials and Methods:

  • Lines 591-592. Write the formulas of ammonium and nitrate in the correct form: (NH4+) and (NO3-).
  • Lines 598-599. Add detailed information about the instruments used in this experiment: photometry and atomic absorption spectrometry.
  • Line 612. Add the model, the manufacturer, and the city and country where the instruments were produced: pH meter and EC meter.

Conclusions:

  • The conclusion section is lacking in important information. I suggest adding more results for the main parameters investigated in this work in order to help the reader with a take-home message.

Author Response

We thank the reviewer for the valuable comments. All suggestions were carefully considered and fully incorporated into the revised manuscript.

Reviewer's comment: Lines 32-33. Add some numbers in this section for the main parameters investigated (root nodulation, nitrogen assimilation rate, accumulation of P, K, Ca, and Mg in plant tissues) as a percentage of increase/decrease.

Answer: The requested numerical values for root nodulation, nitrogen assimilation, and accumulation of P, K, Ca, and Mg were added to the Abstract as recommended.

Reviewer's Comment: Introduction:

I find this section a little bit short and incomplete. Add more information about the importance of legumes in the human diet, their economic importance worldwide, main producers, etc., supported by the literature.  

Answer: Additional information on the nutritional relevance of legumes, their global economic importance, major producing regions, and updated literature references have been incorporated into the Introduction as recommended.

Reviewer's comment: Results:

Figure 3 and Figure 4: Indicate if the results are fresh-weight-based or dry-weight-based in the didascaly in order to facilitate the readers' understanding.

Answer: We thank the reviewer for the observation. The indication DM (dry matter) has been added to the y-axes of the figures to clarify that all results are expressed on a dry-matter basis.

Reviewer's comment: Discussions:

Lines 436-460. Multivariate Analysis of Plant Traits (PCA). The discussion of the PCA section is very short and incomplete, in my opinion. The authors should explain in detail the finding point by point and not give only some general argumentations.

Answer: The authors agree with this important observation. Therefore, sentences discussing the effects of nutrient absorption on different plant structures and genotypes were inserted, with their respective references, emphasizing the associated physiological and growth effects. The authors acknowledge that these additions significantly improved the quality of the text.

Reviewer's comment: Materials and methods:

Lines 591-592. Write the formulas of ammonium and nitrate in the correct form: (NH4+) and (NO3-).

Lines 598-599. Add detailed information about the instruments used in this experiment: photometry and atomic absorption spectrometry.

Line 612. Add the model, the manufacturer, and the city and country where the instruments were produced: pH meter and EC meter.

Answer:

The requested corrections have been implemented. The ionic formulas were revised, additional details about the photometry and atomic absorption instruments were included, and the model, manufacturer, and origin of the pH and EC meters were added.

Reviewer's comment: Conclusions: The conclusion section is lacking in important information. I suggest adding more results for the main parameters investigated in this work in order to help the reader with a take-home message.

Answer: The Conclusions section has been revised accordingly, incorporating additional key findings and addressing the main parameters investigated in the study. The revisions were made in line with this comment and in conjunction with the responses to the other reviewers, with the aim of clarifying the main message and strengthening the overall interpretation of the results.

Reviewer 3 Report

Comments and Suggestions for Authors

The study aims to compare plant performance and soil dynamics following the inoculation of four bean varieties with two different microbial strain, finding that inoculation enhances plant performance regardless the bean variety.

The introduction offers sufficient background to understand the framework of the study. However, although the authors outline the objectives at the end of this section, explicit scientific hypotheses are missing. Since this is an experimental study, formulating clear hypotheses is essential.

Lines 89-93 are not needed in the results section. The information they contain is indicated in M&M.

Figures, in general, are difficult to follow. Could they been broken into different figures thus to make each graph larger?. Fewer colors would also aid the reader in analysing the data.

Line 98: When they say that there was a substantial increase, do they mean a statistically significant increase in whatever the variable?

I’m not sure that the PCA analysis is really needed. This technique is generally used to reduce the dimension of a very high number of variables. In this study, the number of variables is rather limited, as is the number of species under consideration. If this analysis were removed, the manuscript would still retain value in a more concise fashion, which would add clarity to this rather long manuscript.

Lines 338-342: the manuscript says ‘A. brasilense acts mainly through indirect mechanisms, including phytohormone synthesis, modulation of root architecture, and alteration of nutrient availability, which can promote rhizobial infection and the formation of functional nodules…’ How can the authors make this claim if no data on phytohormone production or root analyses were presented? This assertion is speculative and cannot be inferred from the study. They can only claim that the inoculated roots produced more biomass and nodules, and appeared more intricated, but no supporting data is provided. Similarly, the nodules content was not analysed, so it is not possible to assert whether A. brasiliense played any role in them.

Lines 344-350: The authors present a very interesting difference in N distribution in plants associated with the varieties, but they do not discuss why this is happening. This finding could have agronomic consequences and in my opinion, it should be explored in depth. The authors attribute the distinct distribution of N in plant parts to the inoculation treatment and hold the inoculated microorganisms solely responsible for the plant responses. However, in the control treatment there was no inoculation, yet the plants formed nodules. This can only be due to either the presence of nodule-forming bacteria in the soil or contamination from adjacent treatments. Again, the claims are rather speculative, as we do not know what was inside the nodules.

This line of speculation continues throughout the discussion and needs to be polished. Another example is in lines 394–396, where the authors state that the effect on P metabolism due to the microorganisms “Such effects increase P availability for phosphorylation reactions, ATP synthesis, and ribulose-1,5-bisphosphate regeneration, thereby supporting photosynthesis and vegetative growth [48–50].” This has not been proven in the study. While this information is useful in the discussion, it is not appropriately presented. The entire part of the discussion devoted to plant nutrition and growth needs revision to eliminate information not supported by the results.

The section devoted to PCA, as mentioned above, does not really contribute to the study. The manuscript is overly extensive and needs to be shortened. This could be an ideal candidate for reduction.

Line 530: Where was the soil collected from?

Line 540: They used only four plants per treatment, which seems too small a sample size to draw general conclusions.

The current Conclusions section simply summarizes the results. The conclusions must address the scientific hypotheses and explain whether these were supported. This section needs to be completely rewritten in light of the new hypotheses posed.

Author Response

Reviewer's comment: The introduction offers sufficient background to understand the framework of the study. However, although the authors outline the objectives at the end of this section, explicit scientific hypotheses are missing. Since this is an experimental study, formulating clear hypotheses is essential.

Answer: The authors once again express their gratitude for the important observation. In this new version of the manuscript, we have added the study's main hypotheses to the end of the introduction.

Reviewer's comment: Lines 89-93 are not needed in the results section. The information they contain is indicated in M&M.

Answer: The authors appreciate the observation. Part of the paragraph was removed because it had already been presented in the materials and methods, leaving only the sentence directly related to the results: “Factorial ANOVA revealed significant effects of the variety and inoculation treatment on the number of nodules and nodule dry mass.”

Editor's comment: Figures, in general, are difficult to follow. Could they been broken into different figures thus to make each graph larger?. Fewer colors would also aid the reader in analysing the data.

Answer: We agree with the suggestion and have reorganized the figures. The original composite figures were split into separate ones, allowing for significantly larger graphs and improved readability. Regarding the colors, with all due respect and with the editor's and reviewers' permission, we have opted to maintain the current scheme, as we believe it is essential to clearly distinguish the four inoculation treatments. This consistency across all figures allows the reader to quickly track and compare each specific treatment throughout the manuscript.

Reviewer's comment: Line 98: When they say that there was a substantial increase, do they mean a statistically significant increase in whatever the variable?

Answer: Thank you for your comment. In the text, when we use "substantial increase," it is to indicate statistically significant increases according to the statistical analysis adopted in the parameters evaluated in relation to the number of nodules and dry biomass of nodules of the inoculated treatments when compared to the non-inoculated control. We revised the manuscript to make it easier to understand the terms used in the description of the results and what is presented in the figures and statistical analysis.

Reviewer's comment: I’m not sure that the PCA analysis is really needed. This technique is generally used to reduce the dimension of a very high number of variables. In this study, the number of variables is rather limited, as is the number of species under consideration. If this analysis were removed, the manuscript would still retain value in a more concise fashion, which would add clarity to this rather long manuscript.

Answer: The authors agree that PCA is a widely used statistical tool for analyzing high-dimensional data, as it summarizes characteristics by reducing the dimensionality of the original dataset. In fact, it preserves the total variance of the sample but redistributes it among new axes, called principal components. Furthermore, PCA handles multicollinearity (highly correlated characteristics). If a small dataset exhibits strong linear relationships between variables, PCA can effectively reduce dimensionality while preserving most of the variance. Thus, PCA can be used to visualize small datasets by reducing them to two or three dimensions, allowing for easier interpretation and exploration of data trends while preserving the maximum amount of variance in the data, and therefore being more suitable for identifying the most important data variables. In this way, PCA can capture essential patterns even with a limited number of observations (samples).

Reviewer's comment: Lines 338-342: the manuscript says ‘A. brasilense acts mainly through indirect mechanisms, including phytohormone synthesis, modulation of root architecture, and alteration of nutrient availability, which can promote rhizobial infection and the formation of functional nodules…’ How can the authors make this claim if no data on phytohormone production or root analyses were presented? This assertion is speculative and cannot be inferred from the study. They can only claim that the inoculated roots produced more biomass and nodules, and appeared more intricated, but no supporting data is provided. Similarly, the nodules content was not analysed, so it is not possible to assert whether A. brasiliense played any role in them.

Answer: The authors appreciate the reviewer’s feedback. Following a thorough evaluation, the authors conclude that the data do not substantiate this claim. Accordingly, the sentence has been removed from the manuscript. The authors thank the reviewer for this valuable observation.

Reviewer's comment: Lines 344-350: The authors present a very interesting difference in N distribution in plants associated with the varieties, but they do not discuss why this is happening. This finding could have agronomic consequences and in my opinion, it should be explored in depth. The authors attribute the distinct distribution of N in plant parts to the inoculation treatment and hold the inoculated microorganisms solely responsible for the plant responses. However, in the control treatment there was no inoculation, yet the plants formed nodules. This can only be due to either the presence of nodule-forming bacteria in the soil or contamination from adjacent treatments. Again, the claims are rather speculative, as we do not know what was inside the nodules.

Answer: We thank the reviewer for the comment, as the differentiated distribution of nitrogen among the varieties deserves a more in-depth discussion. In the revised version, we expanded the interpretation of the results by highlighting that these differences may be attributed not only to the inoculated microorganisms but also to the intrinsic genetic control of the varieties, which regulates traits such as nodulation efficiency. Regarding the presence of nodules in the control treatment, we have now included an explanation in the manuscript acknowledging that the observed nodulation may result from the role of mineral nitrogen in the plant regulatory system, activating nitrate signaling pathways that lead to the inhibition of genes associated with nodulation, such as NIN and ENOD40.

Reviewer's comment: This line of speculation continues throughout the discussion and needs to be polished. Another example is in lines 394–396, where the authors state that the effect on P metabolism due to the microorganisms “Such effects increase P availability for phosphorylation reactions, ATP synthesis, and ribulose-1,5-bisphosphate regeneration, thereby supporting photosynthesis and vegetative growth [48–50].” This has not been proven in the study. While this information is useful in the discussion, it is not appropriately presented. The entire part of the discussion devoted to plant nutrition and growth needs revision to eliminate information not supported by the results.

Answer: We agree with the reviewer's comments. The discussion section regarding plant nutrition and growth has been thoroughly revised, and all statements not directly supported by our results have been removed or rewritten to align with the data presented.

Reviewer's comment: The section devoted to PCA, as mentioned above, does not really contribute to the study. The manuscript is overly extensive and needs to be shortened. This could be an ideal candidate for reduction.

Answer: We thank the Reviewer for this suggestion. The PCA section has been revised and streamlined based on the comments raised by the other two reviewers, among a total of three reviewers for this manuscript, aiming to improve clarity and relevance while reducing redundancy. However, if the Reviewer still considers that this section does not add sufficient value and recommends its removal, we are fully willing to exclude it from the manuscript.

Reviewer's comment: Line 530: Where was the soil collected from?

Answer: We thank the Reviewer for this comment. The information regarding the soil collection site has now been added to the text.

Reviewer's comment: Line 540: They used only four plants per treatment, which seems too small a sample size to draw general conclusions.

Answer: We thank the Reviewer for this observation. The experiment was conducted with four plants per treatment, following a design commonly adopted in controlled greenhouse studies involving nodulation and inoculation with plant growth-promoting bacteria, where each plant is considered an independent experimental unit. Under protected conditions, environmental variability is substantially reduced, allowing reliable detection of treatment effects even with a limited number of replicates. In addition, the completely randomized 4 × 4 factorial design enabled the evaluation of main effects and interactions between varieties and inoculation treatments using appropriate statistical analyses. The robustness of the experimental design is supported by the significant effects observed and by acceptable coefficients of variation obtained for the evaluated parameters.

Reviewer's comment: The current Conclusions section simply summarizes the results. The conclusions must address the scientific hypotheses and explain whether these were supported. This section needs to be completely rewritten in light of the new hypotheses posed.

Answer: We thank the Reviewer for this comment. The Conclusions section has been completely rewritten to explicitly address the scientific hypotheses of the study and to clarify whether they were supported by the results, incorporating the revised hypotheses and the main findings in an integrative and interpretative manner.

Round 2

Reviewer 1 Report

Comments and Suggestions for Authors

The author responded well to the comments and improved the manuscript.

Author Response

Dear Reviewer,

Thank you for your positive comments. We would like to inform you that a revised version has been submitted, incorporating additional suggestions provided by another reviewer.

Reviewer 2 Report

Comments and Suggestions for Authors

The authors have correctly responded to all of the comments and suggestions.

I recommend accepting the manuscript in its present form.

Author Response

(The authors gave the same response as above.)

Reviewer 3 Report

Comments and Suggestions for Authors

I do acknowledge the effort made by the authors to incorporate all my suggestions into the manuscript. I believe that it has increased it quality and that ir reads better now.

In regards the PCA analysis, I still consider that they do not contribute to the study. On contrary, they add extre repeated information that do not help the reader in understandign the actual meaning of the study. I do insist that the manuscript would benefict by removing this part and simplyfing it a bit. 

Author Response

Dear Reviewer,

We appreciate your careful evaluation and constructive feedback. Following your suggestion, we have removed the PCA analysis from the manuscript. We believe this change significantly improved the flow and focus of our study.